# No association between the vitamin D-binding protein (*DBP*) gene polymorphisms (rs7041 and rs4588) and multiple sclerosis and type 1 diabetes mellitus: A meta-analysis

Xin Zhang[1], Bai Gao[2], Bing Xu[1]*

1 Department of Neurology, Shenyang First People's Hospital, Shenyang, Liaoning Province, China,
2 Department of Nerve Function, Shengjing Hospital of China Medical University, Shenyang, Liaoning Province, China

* xb1968131@163.com

**Data Availability Statement:** All relevant data are within the manuscript and its Supporting information files.

**Funding:** The authors received no specific funding for this work.

## Abstract

### Background

The association between polymorphisms in vitamin D-binding protein (*DBP*) gene and the risk of multiple sclerosis (MS) and type 1 diabetes mellitus (T1DM) has been investigated in many studies, but the studies showed controversial results. The rationale for this meta-analysis was to determine whether *DBP* polymorphisms increases the risk of MS and T1DM by pooling data.

### Methods

Potentially relevant studies were searched using GWAS Catalog, PubMed, Embase, CNKI and WANFANG databases up to November 2019. The pooled odds ratios (ORs) and corresponding 95% confidence intervals (CIs) were performed to estimate the associations in a fixed-effects or random-effects model.

### Results

A total of 13 studies were enrolled in this meta-analysis, including eight studies for MS and five for T1DM. The overall results showed that there was no significant association of *DBP* rs7041 and rs4588 polymorphisms with the risk of MS and T1DM under any genetic model. Similarly, subgroup analysis by ethnicity revealed that no significant association of rs7041 and rs4588 polymorphisms with the risk of MS and T1DM was observed in white or non-white racial groups.

### Conclusions

This meta-analysis provides evidence that *DBP* rs7041 and rs4588 polymorphisms may not be associated with an increased risk in MS and T1DM. However, these findings need further validation by larger-scale epidemiological studies and genome-wide association studies (GWASs) in different populations.

**Competing interests:** The authors have declared that no competing interests exist.

**Abbreviations:** CIs, confidence intervals; CNS, central nervous system; DBP, vitamin D-binding protein; GC, group-specific component; GWASs, genome-wide association studies; HWE, Hardy-Weinberg equilibrium; MESH, medical subjective headings; MS, multiple sclerosis; NOS, Newcastle-Ottawa Scale; ORs, odds ratios; PRISMA, Systematic Reviews and Meta-Analyses; SNPs, single nucleotide polymorphisms; T1DM, type 1 diabetes mellitus; Th17, T helper 17.

## Introduction

Autoimmune diseases are multifactorial with combination of genetic susceptibility and environmental factors leading to the etiology of diseases [1, 2]. The hypothesis that these diseases may share common genetic susceptibility loci has been supported by the epidemiological observation of co-occurrence of multiple autoimmune diseases within a single family [3, 4]. Multiple sclerosis (MS) and Type 1 diabetes mellitus (T1DM) are classified as autoimmune diseases caused by immune response against self-antigens, which lead to immune-mediated damage of self-tissues and organs [5, 6]. MS is an autoimmune demyelinating disease of the central nervous system (CNS) [5]. T1DM is a metabolic disease characterized by absolute insulin deficiency resulting from autoimmune destruction of pancreatic β-cell that produce insulin [6]. Although the etiologies of these autoimmune diseases are not well understood, complex interactions between the environmental triggers and genetic factors have been determined.

The serum vitamin D-binding protein (DBP, group-specific component, GC) is known to function as an immunomodulatory factor, as well as the important carrier protein for vitamin D and its biologically active metabolite, 1,25(OH)2D3 [7]. It has now been shown that vitamin D has a wide range of immune actions [8, 9]. The DBP binds to vitamin D and its metabolites and transports them to various target tissues [10]. When the *DBP* gene mutations occur, serum vitamin D level will be decreased though the patient having sufficient sun exposure or vitamin D supplement. Two single nucleotide polymorphisms (SNPs) of the *DBP* gene, rs7041 and rs4588, have been widely studied [11–13]. GAT→GAG substitution at rs7041 leads to aspartic acid to glutamic acid and ACG→AAG substitution at rs4588 leads to changes of amino acid threonine for Lysine. These two mutations have lower binding capacity for vitamin D [14], and vitamin D deficiency can be linked to the pathophysiology of autoimmune diseases [15]. Although the precise etiology of MS remains unclear, the subset of CD4+ T cells, T helper 17 (Th17) cells have been proposed to play a significant role in inflammatory response [16]. Deficiency of vitamin D and 1,25(OH)2D3 fails to inhibit Th17 cells proliferation, which produce IL-17, TNF-α. Macrophages into the CNS ultimately lead to the pathogenesis of MS [17]. T1DM results from the destruction of insulin-producing β-cells within the islets of Langerhans in the pancreas [18]. 1,25(OH)2D3 inhibits antigen-induced T-cell proliferation and Th1-associated cytokine production, which regulate CD8+ lymphocytes and macrophages [19]. Macrophages and CD8+ T cells infiltrates the islets' interstitium [20], leading to β-cells loss [21]. Insufficient insulin production results in impaired blood glucose regulation, which ultimately causes T1DM.

Up to now, the association between *DBP* rs7041 and rs4588 polymorphisms and risk of MS and T1DM has been reported in several studies, whereas the results remains controversial. In addition, the association between rs7041 and rs4588 *DBP* polymorphisms and MS and T1DM risk has not been included in genome-wide association studies (GWASs) or meta-analysis studies. The rationale for this meta-analysis was to determine whether *DBP* polymorphisms increases the risk of MS and T1DM by pooling data. The meta-analysis was a powerful way to effectively increase the sample size to provide a more valid pooled estimate. Therefore, we performed the comprehensive meta-analysis to derive a more precise estimations of the association between *DBP* polymorphism and the risk of MS and T1DM.

## Methods

This meta-analysis was conducted and reported according to the Preferred Reporting Items for Systematic Reviews and Meta-Analyses (PRISMA) 2009 checklist (S1 Checklist) [22].

## Search strategy

A comprehensive literature search was performed in GWAS Catalog, PubMed, Embase, CNKI and WANFANG databases to explore eligible literature up to November 2019. Terms "diabetes mellitus" or "multiple sclerosis", "rs7041" or "rs4588" were used in GWAS Catalog. And in other databases, the combination of medical subjective headings (MESH) and text words were as follows: "polymorphism, genetic" or "polymorphism" or "polymorphisms" or "genome-wide association study" or "genome-wide association studies" or "GWAS" or "rs7041" or "rs4588", "vitamin D-binding protein" or "DBP" or "group-specific component" or "GC", "multiple sclerosis" or "diabetes mellitus". The literature language was limited to English or Chinese and subjects were limited to humans. References from these papers and reviews were manually retrieved for additional eligible studies.

## Inclusion and exclusion criteria

Inclusion criteria were: (1) studies on the associations between *DBP* rs7041 or rs4588 polymorphism and MS or T1DM; (2) sufficient data for calculating odds ratios (ORs) with accompanying 95% confidence intervals (CIs); (3) published case-control studies in English or Chinese. Exclusion criteria were: (1) reviews, case reports, meta-analyses, letters, and editorials; (2) studies lacking detailed genotype data; (3) overlapped data.

## Data extraction and quality assessment

Two investigators extracted the following data independently from each study: first author's name, year of publication, ethnicity, the genotype frequencies, the *P*-value of Hardy-Weinberg equilibrium (HWE) of controls, and Newcastle-Ottawa Scale (NOS) score [23]. Discrepancies were resolved by discussion. The NOS was carried out to assess the methodological quality of eligible case-control studies on 3 aspects: selection, comparability, and exposure. The total NOS score ranges from 0 to 9 stars, and study was assumed to be high methodological quality if the score was 6 or more.

## Statistical analysis

A $\chi^2$ test was performed to measured deviation from the HWE in controls. The pooled ORs and 95% CIs were used to estimate the association between the polymorphisms in the *DBP* gene and risk of MS and T1DM. For *DBP* rs7041 polymorphism, the allele (G versus T), dominant (GG+GT versus TT), recessive (GG versus GT+TT), homozygous (GG versus TT), and heterozygous model (GG versus GT) were evaluated. For the *DBP* rs4588 polymorphism, the allele (A versus C), dominant (AA + AC versus CC), recessive (AA versus AC + CC), homozygous (AA versus CC), and heterozygous model (AA versus AC) were evaluated. Heterogeneity between studies was determined using $\chi^2$ based *Q*-test and quantified using the $I^2$ statistics [24]. When *P* value> 0.1 or $I^2$<50%, pooled ORs were calculated by fixed-effects model (Mantel-Haenszel method). Otherwise, the random-effects model (DerSimonian and Laird method) was applied [25]. Furthermore, subgroup analyses were conducted based on ethnicity (white and non-white) to obtain the sources of heterogeneity. The white group included Caucasian and European, and the non-white group consisted of Hispanic, Black, Asian and Bengali. To determine potential publication bias, the Egger's linear regression test was used for overall genetic models [26]. If there was publication bias, we recalculated the adjusted ORs using the trim-and-fill method [27] to evaluate the possible impact of publication bias. Sensitivity analyses were carried out by removing studies deviated from HWE to assess the robustness of the results. The 2-tailed *P* value<0.05 was considered statistically significant. Statistical analyses

were performed using Stata 15.0 software (Stata Corporation, College Station, TX, USA). Moreover, statistical power analyses were conducted using G*Power software (version 3.1.9.2) [28] to determine whether the meta-analysis had sufficient power ($\geq$80%).

## Results

### Literature research and study characteristics

After a comprehensive literature search, 901 articles were initially identified, and 140 duplicate articles were removed. After checking titles and abstracts, 738 articles were excluded. A further 12 articles were excluded after checking the full text. Overall, a total of 11 articles (consisting of 13 studies) met our inclusion criteria were enrolled in our meta-analysis [29–39]. Of these 13 studies: 8 studies (6 articles) assessed the association of *DBP* gene polymorphisms with MS risk [29–34] and 5 studies (5 articles) investigated the association between *DBP* gene polymorphisms and risk of T1DM [35–39]. Furthermore, there were no GWASs and meta-analyses published in this area. GWAS catalog search yielded 805 and 323 results for the term "multiple sclerosis" and "type I diabetes mellitus" respectively. However, none of them presented results of *DBP* polymorphisms examined in genetic association studies. The NOS scores of studies ranged from 3 to 8 stars. The characteristics and NOS scores of included studies are shown in Table 1. Fig 1 showed the detailed screening process for the articles involved in the meta-analysis.

### *DBP* rs7041 and rs4588 polymorphism and MS risk

To determine the association of *DBP* gene polymorphism with the MS risk, 8 studies (involving 1600 cases and 1770 controls) about *DBP* rs7041 polymorphism and 8 studies (involving 1600 cases and 1771 controls) about *DBP* rs4588 polymorphism were enrolled in our meta-analysis. As shown in Table 2, no association was observed between *DBP* rs7041 and rs4588 gene polymorphisms and risk of MS in overall population under any genetic model. Furthermore, stratification by ethnicity failed to explore any association of these polymorphisms with risk of MS in the white and non-white racial group.

### *DBP* rs7041 and rs4588 polymorphism and T1DM risk

To determine the potential association between *DBP* gene polymorphism and the risk of T1DM, 5 studies (involving 1843 cases and 2151 controls) about *DBP* rs7041 polymorphism and 4 studies (involving 1712 cases and 2056 controls) about *DBP* rs4588 polymorphism were enrolled in this meta-analysis. As shown in Table 2, no association was found between polymorphisms (rs7041 and rs4588) in the *DBP* gene and risk of T1DM in overall population under overall genetic models. Furthermore, stratification by ethnicity failed to explore any association between these polymorphisms and risk of T1DM in the white and non-white racial group.

### Publication bias

The Egger's linear regression test was conducted to detect the publication biases under all genetic models. For *DBP* rs7041 polymorphism and MS risk, the results indicated that there was evidence for publication bias in the dominant model (GG+GT vs. TT: *P* = 0.034). The trim and fill method revealed no difference between the unadjusted (OR = 0.93, 95% CI 0.77–1.13, *P* = 0.034) and adjusted (OR = 0.91, 95% CI 0.75–1.10, *P* = 0.322) results, which indicated that the stability of the results was not affected by the presence of publication bias. For *DBP*

**Table 1. Characteristics of the included studies in this meta-analysis.**

| Author | Year | Ethnicity | Sample size | | Genotype distribution | | | | | | HWE | NOS |
|---|---|---|---|---|---|---|---|---|---|---|---|---|
| | | | Case | Control | Case | | | Control | | | P value | score |
| **rs7041 and MS** | | | | | GG | GT | TT | GG | GT | TT | | |
| Langer-Gould A | 2018 | White | 247 | 267 | 122 | 105 | 20 | 141 | 99 | 27 | 0.13 | 5 |
| Langer-Gould A | 2018 | Hispanic | 183 | 197 | 110 | 66 | 7 | 112 | 75 | 10 | 0.57 | 5 |
| Langer-Gould A | 2018 | Black | 116 | 131 | 96 | 18 | 2 | 98 | 29 | 4 | 0.32 | 5 |
| Agliardi C | 2017 | White | 701 | 831 | 226 | 350 | 125 | 285 | 412 | 134 | 0.46 | 6 |
| Agnello L | 2017 | White | 100 | 92 | 36 | 41 | 23 | 37 | 35 | 20 | 0.04 | 6 |
| Simon KC | 2010 | White | 100 | 100 | 33 | 51 | 16 | 30 | 51 | 19 | 0.75 | 5 |
| Li XH | 2010 | Asian | 46 | 43 | 0 | 16 | 30 | 0 | 21 | 22 | 0.03 | 5 |
| Niino M | 2002 | Asian | 107 | 109 | 4 | 31 | 72 | 6 | 36 | 67 | 0.69 | 5 |
| **rs4588 and MS** | | | | | AA | AC | CC | AA | AC | CC | | |
| Langer-Gould A | 2018 | White | 247 | 267 | 56 | 116 | 75 | 52 | 123 | 92 | 0.35 | 5 |
| Langer-Gould A | 2018 | Hispanic | 183 | 197 | 40 | 93 | 50 | 47 | 100 | 50 | 0.83 | 5 |
| Langer-Gould A | 2018 | Black | 116 | 131 | 79 | 32 | 5 | 94 | 31 | 6 | 0.12 | 5 |
| Agliardi C | 2017 | White | 701 | 831 | 43 | 276 | 382 | 54 | 331 | 446 | 0.48 | 6 |
| Agnello L | 2017 | White | 100 | 92 | 7 | 32 | 61 | 8 | 27 | 57 | 0.08 | 6 |
| Simon KC | 2010 | White | 100 | 101 | 5 | 41 | 54 | 9 | 39 | 53 | 0.64 | 5 |
| Li XH | 2010 | Asian | 46 | 43 | 4 | 25 | 17 | 2 | 29 | 12 | <0.05 | 5 |
| Niino M | 2002 | Asian | 107 | 109 | 3 | 49 | 55 | 6 | 45 | 58 | 0.47 | 5 |
| **rs7041 and T1DM** | | | | | GG | GT | TT | GG | GT | TT | | |
| Kirac D | 2018 | White | 55 | 40 | 8 | 27 | 20 | 5 | 15 | 20 | 0.42 | 6 |
| Blanton D | 2011 | White | 1454 | 1828 | 441 | 723 | 290 | 579 | 884 | 365 | 0.41 | 3 |
| Ongagna JC | 2005 | White | 110 | 68 | 41 | 53 | 16 | 12 | 39 | 17 | 0.21 | 8 |
| Ongagna JC | 2001 | White | 43 | 52 | 16 | 21 | 6 | 6 | 22 | 24 | 0.78 | 5 |
| Klupa T | 1999 | White | 181 | 163 | 60 | 80 | 41 | 46 | 85 | 32 | 0.52 | 6 |
| **rs4588 and T1DM** | | | | | AA | AC | CC | AA | AC | CC | | |
| Kirac D | 2018 | White | 55 | 40 | 3 | 20 | 32 | 1 | 12 | 27 | 0.81 | 6 |
| Blanton D | 2011 | White | 1433 | 1801 | 132 | 578 | 723 | 138 | 734 | 929 | 0.67 | 3 |
| Ongagna JC | 2001 | White | 43 | 52 | 0 | 14 | 29 | 2 | 16 | 34 | 0.95 | 5 |
| Klupa T | 1999 | White | 181 | 163 | 10 | 76 | 95 | 16 | 65 | 82 | 0.56 | 6 |

Abbreviations: MS, multiple sclerosis; T1DM, type 1 diabetes mellitus; HWE, Hardy-Weinberg equilibrium; NOS, Newcastle-Ottawa scale.

rs4588 polymorphism and MS risk, and *DBP* rs7041 or rs4588 polymorphism and T1DM risk, no evidence of publication bias was found.

## Sensitivity analysis

The genotype distribution in the control subjects in two studies [30, 34] significantly deviated from the HWE in our meta-analysis. Sensitivity analysis was carried out by omitting the HWE-violating studies to estimate the robustness of our conclusions, and the results were materially changed after excluding the two studies (Table 2).

## Statistical power analysis

Statistical power analyses were calculated to detect the powers of the association of the *DBP* rs7041 and rs4588 polymorphism with the risk of MS and T1DM with $\alpha = 0.05$ and $\beta = 0.2$.

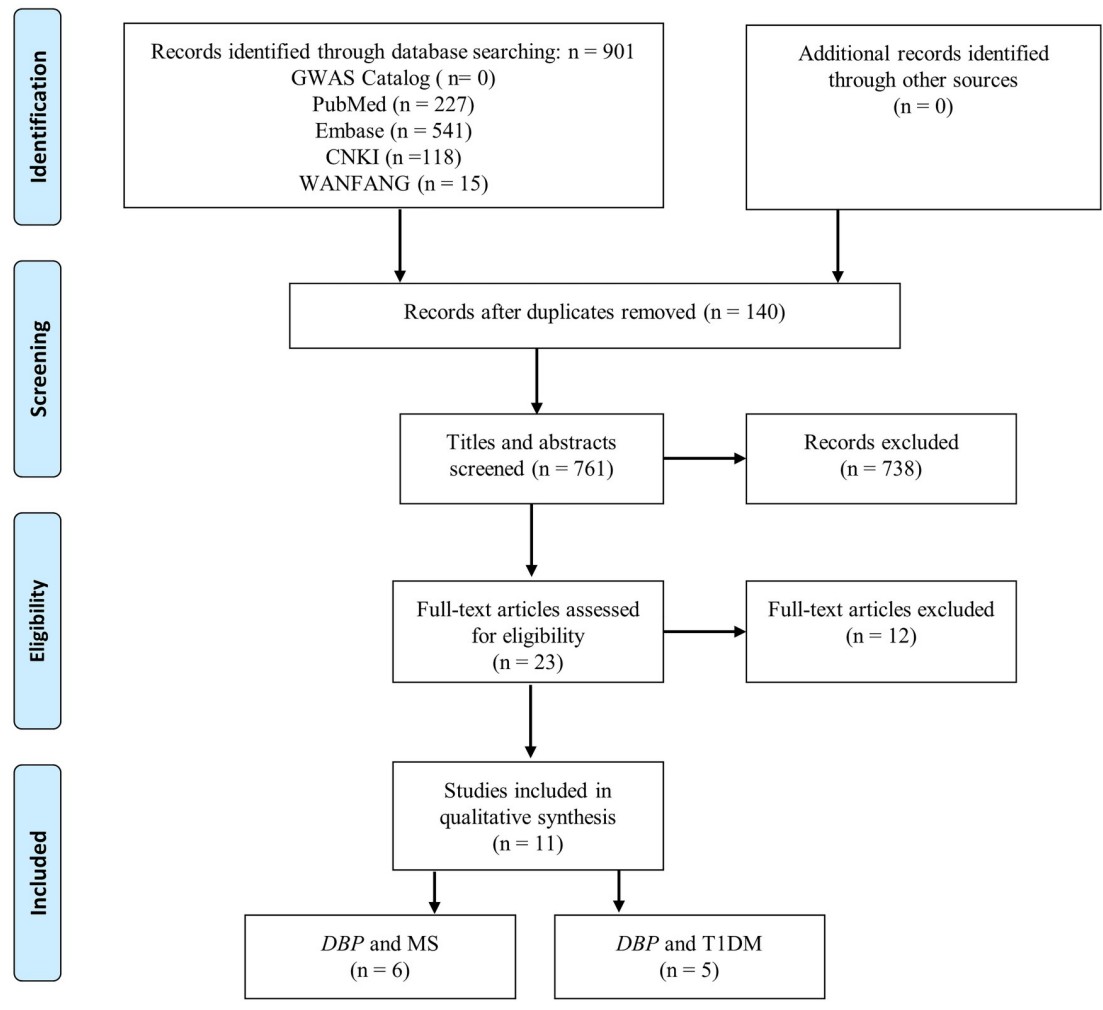

**Fig 1. PRISMA flow diagram.**

The power of each pooled result ranged from 5 to 17%, and the specific power values are summarized in Table 2.

## Discussion

Our study demonstrated that there not be any statistically significant difference for rs7041 or rs4588 alleles and genotypes of *DBP* gene between MS or T1DM and control.

There should be differences between Asian and the other ethnicities in non-white racial groups. Unfortunately, there were insufficient number of studies in different ethnicities. For *DBP* polymorphism and MS risk, 1 study involving Hispanic, 1 study involving Black, 2 studies involving Asian, and 4 studies involving white were enrolled. For *DBP* polymorphism and T1DM risk, 5 studies involving white were enrolled. Because of insufficient number of studies in different ethnicities, study population was categorized as being white and non-white racial groups on basis of ethnicity in this study. The former group included Caucasian and European, and the latter group consisted of Hispanic, Black, Asian and Bengali. The findings of our meta-analysis should be interpreted with caution in the case of limited number of studies, more studies on different ethnicities are needed in the future.

**Table 2. Results of meta-analysis for *DBP* polymorphism and MS and T1DM.**

| SNP | Population | Studies (n) | Test of heterogeneity | | Test of associations | | $P_{Egger}$ | Power analysis (%) | $P_{Sensitivity}$ |
|---|---|---|---|---|---|---|---|---|---|
| | | | P value | $I^2$(%) | OR (95% CI) | P value | | | |
| **rs7041 and MS** | | | | | | | | | |
| G vs. T | Overall | 8 | 0.464 | 0.0 | 0.96 (0.87, 1.07) | 0.493 | 0.440 | 15 | 0.689 |
| | White | 4 | 0.810 | 0.0 | 0.95 (0.84, 1.06) | 0.352 | 0.455 | 14 | 0.418 |
| GG vs. TT | Overall | 7 | 0.774 | 0.0 | 0.95 (0.75, 1.19) | 0.650 | 0.138 | 17 | 0.735 |
| | White | 4 | 0.669 | 0.0 | 0.93 (0.73, 1.18) | 0.532 | 0.271 | 11 | 0.609 |
| GG vs. GT | Overall | 7 | 0.675 | 0.0 | 0.96 (0.82, 1.13) | 0.643 | 0.908 | 5 | 0.737 |
| | White | 4 | 0.845 | 0.0 | 0.91 (0.76, 1.08) | 0.286 | 0.978 | 15 | 0.344 |
| GG+GT vs. TT | Overall | 8 | 0.706 | 0.0 | 0.93 (0.77, 1.13) | 0.474 | 0.034 | 17 | 0.708 |
| | White | 4 | 0.648 | 0.0 | 0.96 (0.77, 1.20) | 0.736 | 0.210 | 7 | 0.776 |
| GG vs. GT+TT | Overall | 7 | 0.573 | 0.0 | 0.97 (0.83, 1.12) | 0.666 | 0.790 | 8 | 0.774 |
| | White | 4 | 0.867 | 0.0 | 0.91 (0.77, 1.08) | 0.280 | 0.706 | 16 | 0.343 |
| **rs4588 and MS** | | | | | | | | | |
| A vs. C | Overall | 8 | 0.928 | 0.0 | 0.98 (0.88, 1.09) | 0.760 | 0.745 | 6 | 0.803 |
| | White | 4 | 0.595 | 0.0 | 1.01 (0.89, 1.14) | 0.911 | 0.911 | 5 | 0.911 |
| AA vs. CC | Overall | 8 | 0.814 | 0.0 | 0.96 (0.75, 1.24) | 0.776 | 0.687 | 6 | 0.735 |
| | White | 4 | 0.464 | 0.0 | 1.01 (0.75, 1.36) | 0.941 | 0.393 | 5 | 0.941 |
| AA vs. AC | Overall | 8 | 0.782 | 0.0 | 0.93 (0.74, 1.17) | 0.535 | 0.067 | 8 | 0.451 |
| | White | 4 | 0.628 | 0.0 | 0.97 (0.73, 1.30) | 0.854 | 0.158 | 5 | 0.854 |
| AA+AC vs. CC | Overall | 8 | 0.946 | 0.0 | 1.00 (0.86, 1.15) | 0.947 | 0.872 | 5 | 0.934 |
| | White | 4 | 0.771 | 0.0 | 1.01 (0.86, 1.19) | 0.881 | 0.647 | 5 | 0.881 |
| AA vs. AC+CC | Overall | 8 | 0.770 | 0.0 | 0.94 (0.76, 1.17) | 0.603 | 0.259 | 7 | 0.533 |
| | White | 4 | 0.528 | 0.0 | 1.00 (0.76, 1.32) | 0.993 | 0.236 | 5 | 0.993 |
| **rs7041 and T1DM** | | | | | | | | | |
| G vs. T | Overall (White) | 5 | <0.001 | 83.2 | 1.04 (0.95, 1.14) | 0.345 | 0.070 | 11 | 0.345 |
| GG vs. TT | Overall (White) | 5 | <0.001 | 80.2 | 1.08 (0.91, 1.30) | 0.375 | 0.095 | 12 | 0.375 |
| GG vs. GT | Overall (White) | 5 | 0.023 | 64.7 | 1.02 (0.89, 1.18) | 0.760 | 0.110 | 5 | 0.760 |
| GG+GT vs. TT | Overall (White) | 5 | 0.006 | 72.7 | 1.08 (0.93, 1.26) | 0.318 | 0.144 | 13 | 0.318 |
| GG vs. GT+TT | Overall (White) | 5 | 0.002 | 76.4 | 1.04 (0.91, 1.19) | 0.574 | 0.070 | 7 | 0.574 |
| **rs4588 and T1DM** | | | | | | | | | |
| A vs. C | Overall (White) | 4 | 0.420 | 0.0 | 1.05 (0.95, 1.16) | 0.373 | 0.723 | 12 | 0.373 |
| AA vs. CC | Overall (White) | 4 | 0.185 | 37.8 | 1.14 (0.89, 1.45) | 0.299 | 0.500 | 16 | 0.299 |
| AA vs. AC | Overall (White) | 4 | 0.222 | 31.7 | 1.12 (0.87, 1.43) | 0.373 | 0.402 | 13 | 0.373 |
| AA+AC vs. CC | Overall (White) | 4 | 0.769 | 0.0 | 1.04 (0.91, 1.18) | 0.562 | 0.956 | 8 | 0.562 |
| AA vs. AC+CC | Overall (White) | 4 | 0.179 | 38.9 | 1.13 (0.89, 1.43) | 0.306 | 0.468 | 16 | 0.306 |

Abbreviations: SNP, single nucleotide polymorphism; OR, odds ratio; CI, confidence interval; PEgger, P value for Egger's linear regression test; PSensitivity, P value for sensitivity analysis; MS, multiple sclerosis; T1DM, type 1 diabetes mellitus; NA, not available.

Several limitations involved in our results should be addressed. First, our study included eight articles for MS and five for T1DM, the meta-analysis may be unable to have sufficient power to identify real association because of the limited study number, especially when grouped by ethnicity. Since the power of most genetic models ranged from 5 to 17%, less than 80%, the findings of the meta-analysis may be insufficiently confirmed. Owing to the low statistical power, conclusions drawn from this meta-analysis should be interpreted cautiously. Second, the studies included in this meta-analysis can be organized into two categories according to ethnicity: white and non-white racial groups, however, the latter group consisted of

different ethnicities, including Hispanic, Black, Asian and Bengali. Thus, our conclusions are only applicable to white racial groups, further studies in other ethnicities are needed. Third, selection bias may occur due to the inclusion of only English or Chinese literature. Forth, MS and T1DM are multifactorial disorders resulted from complex interactions between genetic, epigenetic, and environmental factors, suggesting that the *DBP* polymorphism may only partially contribute to the pathogenesis of these chronic diseases, this may lead to bias in our results. Finally, our study only determined the associations between two single loci in the *DBP* gene, rs7041 and rs4588 polymorphisms and the risk of MS and T1DM, but we did not examine associations between *DBP* gene haplotypes and these diseases due to insufficient haplotype data. It remains unclear whether other *DBP* gene mutations can particularly lead to changes in its expression. In terms of the genetic causes of disease, haplotypes can provide more critical information than the corresponding single *SNP*.

In conclusion, our meta-analysis suggests that neither rs7041 nor rs4588 polymorphism of *DBP* is associated with the MS and T1DM risk. Further well-designed studies with larger sample sizes and different ethnicities are needed to validate our conclusion.

## Supporting information

**S1 Checklist. PRISMA 2009 checklist.**
(DOC)

**S2 Checklist. Meta-analysis on genetic association studies checklist.**
(DOCX)

## Acknowledgments

The authors thank all the participants in this study.

## Author Contributions

**Conceptualization:** Bing Xu.

**Data curation:** Xin Zhang.

**Formal analysis:** Xin Zhang.

**Investigation:** Bai Gao.

**Methodology:** Xin Zhang, Bai Gao.

**Project administration:** Bing Xu.

**Resources:** Xin Zhang.

**Software:** Xin Zhang.

**Supervision:** Bing Xu.

**Validation:** Bai Gao.

**Visualization:** Xin Zhang.

**Writing – original draft:** Xin Zhang, Bai Gao, Bing Xu.

**Writing – review & editing:** Bing Xu.

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
