## [Decision Letter · Decision Letter 0]

3 Sep 2020

PONE-D-20-01839

No Association between the vitamin D-binding protein (DBP) gene polymorphisms (rs7041 and rs4588) and multiple sclerosis and type 1 diabetes mellitus: A meta-analysis

PLOS ONE

Dear Dr. Xu,

Thank you for submitting your manuscript to PLOS ONE. After careful consideration, we feel that it has merit but does not fully meet PLOS ONE’s publication criteria as it currently stands. Therefore, we invite you to submit a revised version of the manuscript that addresses the points raised during the review process.

Please review the comments from the reviewer. Specifically, address editorial sections in the paper. 

We look forward to receiving your revised manuscript.

Kind regards,

Joseph Devaney

Academic Editor

PLOS ONE

Journal Requirements:

3.Thank you for stating the following financial disclosure:

 [No].

Reviewers' comments:

Reviewer's Responses to Questions

**Comments to the Author**

1. Is the manuscript technically sound, and do the data support the conclusions?

Reviewer #1: Yes

2. Has the statistical analysis been performed appropriately and rigorously? 

Reviewer #1: Yes

3. Have the authors made all data underlying the findings in their manuscript fully available?

Reviewer #1: Yes

4. Is the manuscript presented in an intelligible fashion and written in standard English?

Reviewer #1: No

5. Review Comments to the Author

Reviewer #1: The paper is a meta-analysis of previous studies of two polymorphisms in the vitamin D binding protein (DBP) in relation to susceptibility to multiple sclerosis and type 1 diabetes. Vitamin D has immunomodulatory and protective effects in autoimmunity, and low vitamin D status appears to be a risk factor for both MS and DM.

Vitamin D is transported by vitamin D binding protein. There are conflicting results in the literature on possible associations between these two SNPs within the DBP gene, and autoimmune disease.

For the meta-analysis, eight studies for MS and five studies for T1DM were selected The methods and statistics appear sound. There is no statistically significant association observed with rs7041 or rs4588 alleles in the pooled results of MS (1843 cases and 2151 controls) or T1DM (1712 cases and 2056 controls).

The authors comment that there was an "insufficient number of studies in different ethnicities". The cohorts were grouped into white (Caucasian and European) and non white (Hispanic Black Asian Bengali) racial groups. All of the non white populations are lumped into a single group. Possible associations may be overlooked in these different populations. In addition, MS and diabetes are both heterogenous conditions, and perhaps there could be an association between DBP polymorphisms and some subtypes of these conditions.

Polymorphisms in DBP affect binding affinity and serum free 25-OH-vitamin D levels, and so affect the amount available to convert to 1,25-OH-vitamin D (Revez, J.A., Lin, T., Qiao, Z. et al. Genome-wide association study identifies 143 loci associated with 25 hydroxyvitamin D concentration. Nat Commun 11, 1647 (2020). https://doi.org/10.1038/s41467-020-15421-7). An association between DBP polymorphisms and diseases such as MS and T1DM might be expected. However there are many other potential factors, such as polymorphisms affecting vitamin D synthesis, catabolism, the vitamin D receptor, and epigenetic effects.

There are several weakness in drawing any broad conclusions, and the authors acknowledge this. However I think the paper is ok within these limitations.

There are a few sections where the grammar or wording could be polished up a bit.

6. PLOS authors have the option to publish the peer review history of their article (what does this mean?). If published, this will include your full peer review and any attached files.

Reviewer #1: No

---

## [Author Response · Author response to Decision Letter 0]

24 Oct 2020

Dear Editor and Reviewer:

Thank you for your letter and for the reviewers’ comments concerning our manuscript entitled “No Association between the vitamin D-binding protein (DBP) gene polymorphisms (rs7041 and rs4588) and multiple sclerosis and type 1 diabetes mellitus: A meta-analysis”(ID: PONE-D-20-01839). Those comments are all valuable and very helpful for revising and improving our paper, as well as the important guiding significance to our research. We have studied comments carefully and have made correction, and we hope to meet with approval. Revised portion are marked in red in the paper. 

Beside，when submitting our revision, we addressed these additional requirements.

3.Thank you for stating the following financial disclosure:

The main corrections in the paper and the responds to the reviewers’ comments are as following:

Reviewers' comments:

Reviewer's Responses to Questions

Comments to the Author

1. Is the manuscript technically sound, and do the data support the conclusions?

Reviewer #1: Yes

2. Has the statistical analysis been performed appropriately and rigorously?

Reviewer #1: Yes

3. Have the authors made all data underlying the findings in their manuscript fully available?

Reviewer #1: Yes

4. Is the manuscript presented in an intelligible fashion and written in standard English?

Reviewer #1: No

Dear Reviewer:

Special thanks to you for your good comments. It is true as reviewer suggested that there are a few sections where the grammar or wording should be polished up a bit. We tried our best to improve the manuscript and made correction in the manuscript. And here we marked in red in revised paper. We appreciate for Reviewer’s warm work earnestly, and hope that the correction will meet with approval. 

5. Review Comments to the Author

Reviewer #1: The paper is a meta-analysis of previous studies of two polymorphisms in the vitamin D binding protein (DBP) in relation to susceptibility to multiple sclerosis and type 1 diabetes. Vitamin D has immunomodulatory and protective effects in autoimmunity, and low vitamin D status appears to be a risk factor for both MS and DM.

Vitamin D is transported by vitamin D binding protein. There are conflicting results in the literature on possible associations between these two SNPs within the DBP gene, and autoimmune disease.

For the meta-analysis, eight studies for MS and five studies for T1DM were selected The methods and statistics appear sound. There is no statistically significant association observed with rs7041 or rs4588 alleles in the pooled results of MS (1843 cases and 2151 controls) or T1DM (1712 cases and 2056 controls).

The authors comment that there was an "insufficient number of studies in different ethnicities". The cohorts were grouped into white (Caucasian and European) and non white (Hispanic Black Asian Bengali) racial groups. All of the non white populations are lumped into a single group. Possible associations may be overlooked in these different populations. In addition, MS and diabetes are both heterogenous conditions, and perhaps there could be an association between DBP polymorphisms and some subtypes of these conditions.

Polymorphisms in DBP affect binding affinity and serum free 25-OH-vitamin D levels, and so affect the amount available to convert to 1,25-OH-vitamin D (Revez, J.A., Lin, T., Qiao, Z. et al. Genome-wide association study identifies 143 loci associated with 25 hydroxyvitamin D concentration. Nat Commun 11, 1647 (2020). https://doi.org/10.1038/s41467-020-15421-7). An association between DBP polymorphisms and diseases such as MS and T1DM might be expected. However there are many other potential factors, such as polymorphisms affecting vitamin D synthesis, catabolism, the vitamin D receptor, and epigenetic effects.

There are several weakness in drawing any broad conclusions, and the authors acknowledge this. However I think the paper is ok within these limitations.

There are a few sections where the grammar or wording could be polished up a bit.

6. PLOS authors have the option to publish the peer review history of their article (what does this mean?). If published, this will include your full peer review and any attached files.

Do you want your identity to be public for this peer review? For information about this choice, including consent withdrawal, please see our Privacy Policy.

Reviewer #1: No

---

## [Editor Report · Decision Letter 1]

30 Oct 2020

No Association between the vitamin D-binding protein (DBP) gene polymorphisms (rs7041 and rs4588) and multiple sclerosis and type 1 diabetes mellitus: A meta-analysis

PONE-D-20-01839R1

Dear Dr. Xu,

We’re pleased to inform you that your manuscript has been judged scientifically suitable for publication and will be formally accepted for publication once it meets all outstanding technical requirements.

Kind regards,

Joseph Devaney

Academic Editor

PLOS ONE
---

## [Editor Report · Acceptance letter]

5 Nov 2020

PONE-D-20-01839R1 

No association between the vitamin D-binding protein (*DBP*) gene polymorphisms (rs7041 and rs4588) and multiple sclerosis and type 1 diabetes mellitus: A meta-analysis 

Dear Dr. Xu:

I'm pleased to inform you that your manuscript has been deemed suitable for publication in PLOS ONE. Congratulations! Your manuscript is now with our production department. 

Kind regards, 

on behalf of

Dr. Joseph Devaney 

Academic Editor

PLOS ONE